# PERK-Mediated eIF2α Phosphorylation Contributes to The Protection of Dopaminergic Neurons from Chronic Heat Stress in *Drosophila*

**DOI:** 10.3390/ijms21030845

**Published:** 2020-01-28

**Authors:** Rosalie Elvira, Sun Joo Cha, Gyeong-Mu Noh, Kiyoung Kim, Jaeseok Han

**Affiliations:** 1Soonchunhyang Institute of Medi-bio Science, Soonchunhyang University, Cheonan, Chungcheongnam-do 31151, Korea; maria.rosalie.elvira@gmail.com (R.E.); cktjswn92@sch.ac.kr (S.J.C.); 2Department of Medical Biotechnology, Soonchunhyang University, Asan, Chungcheongnam-do 31538, Korea; biomedic@sch.ac.kr

**Keywords:** Chronic heat stress, ER stress, eIF2α phosphorylation, PERK, Parkinson disease, *Drosophila*

## Abstract

Environmental high-temperature heat exposure is linked to physiological stress such as disturbed protein homeostasis caused by endoplasmic reticulum (ER) stress. Abnormal proteostasis in neuronal cells is a common pathological factor of Parkinson’s disease (PD). Chronic heat stress is thought to induce neuronal cell death during the onset and progression of PD, but the exact role and mechanism of ER stress and the activation of the unfolded protein response (UPR) remains unclear. Here, we showed that chronic heat exposure induces ER stress mediated by the PKR-like eukaryotic initiation factor 2α kinase (PERK)/eIF2α phosphorylation signaling pathway in *Drosophila* neurons. Chronic heat-induced eIF2α phosphorylation was regulated by PERK activation and required for neuroprotection from chronic heat stress. Moreover, the attenuated protein synthesis by eIF2α phosphorylation was a critical factor for neuronal cell survival during chronic heat stress. We further showed that genetic downregulation of PERK, specifically in dopaminergic (DA) neurons, impaired motor activity and led to DA neuron loss. Therefore, our findings provide in vivo evidence demonstrating that chronic heat exposure may be a critical risk factor in the onset of PD, and eIF2α phosphorylation mediated by PERK may contribute to the protection of DA neurons against chronic heat stress in *Drosophila*.

## 1. Introduction

Many organisms, including humans, could be influenced by the consequences of the increased temperature of the earth, known as global warming. The increase in the rate of neurodegenerative disorders and various cancers, including skin tumors, may be considered a consequence of global warming [1,2,3]. According to the evidence, the brain is a highly susceptible organ to high-temperature heat exposure, in which heat stress can lead to neuronal cell death, cognitive dysfunction, and memory deficits [4,5,6]. Previous studies suggest that heat stress can induce neuronal cell death, monoamine overload, neurological defects, and heat stroke [7,8,9]. Moreover, heat exposure causes disturbed protein homeostasis (proteostasis) in cells [10].

Several neurodegenerative disorders, such as Alzheimer’s disease (AD), Parkinson’s disease (PD), and Huntington’s diseases (HD), share common pathological risk factors, such as abnormal aggregation of misfolded proteins with subsequently disturbed proteostasis. PD is the most frequent neurodegenerative disease that affects older people [11]. Its symptoms involve difficulties in moving or talking, behavioral changes, memory difficulties, rigidity, and trembling, and symptoms usually get worse over time [12]. PD happens as a consequence of dopamine neuron loss in the midbrain substantia nigra. The phenotype of PD involves the aggregation of α-synuclein, which makes Lewy bodies [13]. The cause of PD is mainly environmental factors, such as oxidative stress or toxins. Genetic mutations involved in genes related to mitochondrial dysfunction also have been shown to cause PD [14]. Emerging evidence indicates that hyperthermal stress may be linked to PD pathogenesis. Heat exposure reduces the level of tyrosine hydroxylase (TH) enzyme in rats and impairs dopaminergic (DA) neurons in mice [5]. However, the exact role and mechanism of high-temperature heat exposure in the onset and progression of PD is poorly understood.

The endoplasmic reticulum (ER) is an intracellular organelle with important roles in calcium storage, lipid synthesis, and protein processing. The accumulation of unfolded/misfolded proteins causes severe disturbance in ER homeostasis, referred to as ER stress, and subsequent induction of the unfolded protein response (UPR) [15]. The UPR is mediated by three transmembrane sensor proteins, which are activating transcription factor 6 alpha (ATF6α), inositol-requiring kinase/endoribonuclease 1 alpha (IRE1α), and PKR-like eukaryotic initiation factor 2α kinase (PERK). They are sequestered by the ER chaperone Grp78/BiP under normal conditions. Upon ER stress, GRP78/BiP is dissociated from these transmembrane sensor proteins, which initiates the UPR signaling pathways [16,17]. First, ATF6 is processed in the Golgi into a 50 kDa protein which is transported to the nucleus to serve as a transcription factor [18]. Second, phosphorylation of inositol-requiring kinase/endoribonuclease 1 (IRE1) occurs, which splices X-binding protein 1 (XBP1) into its active transcription factor form. The last is phosphorylation of PERK, which phosphorylates eIF2α and leads to inhibition of global protein translation [19].

Since high-temperature heat exposure causes protein aggregation and denaturation inside cells, it is likely that heat stress could lead to disturbed proteostasis in the ER [20,21]. We previously reported that heat exposure induces ER stress, and the PERK/eIF2α phosphorylation branch of the UPR is essential to protect cells from heat-stress-mediated apoptosis [22]. However, the precise role of ER stress and activation of the UPR signaling pathway by heat exposure in the pathogenesis of neurodegenerative disease has not been explored. ER stress, such as phosphorylation of PERK and eIF2α, has been known to be activated in PD [23]. The role of this eIF2in PD might have positive and negative effects on dopamine neuron survival. Previous studies have suggested that inhibiting the PERK pathway chemically can increase the motor performance in mouse models [24,25]. On the other hand, it was reported that eIF2α activation has a protective effect by reducing protein translation, thus conserving energy resources and enhancing translation of selected mRNA expressions of stress-related protein [26,27,28]. In addition, previous studies have also shown that the downstream of eIF2α phosphorylation ATF4 also protects neuronal cell death in PD models by maintaining Parkin levels [29]. Here, we investigated chronic heat-stress-linked PD pathogenesis mediated by eIF2α phosphorylation using a *Drosophila* model. We found that chronic heat exposure induced ER stress with subsequent induction of the PERK/eIF2α phosphorylation pathway in *Drosophila* neuronal cells. The toxicity induced by chronic heat stress was enhanced in neuron-specific downregulation of PERK. Our results showed that PERK is required for phosphorylation of eIF2α in response to chronic heat-induced ER stress activation and the attenuation of global protein translation that occurs in neuronal cells, including DA neurons. Furthermore, we showed that downregulation of PERK in DA neurons impaired locomotor activity and induced selective loss of DA neurons under chronic heat exposure. Thus, chronic exposure to heat in a *Drosophila* model may hasten the onset and progression of sporadic PD, and chronic heat-induced activation of ER stress mediated by eIF2α phosphorylation suggests a potential pathological mechanism of PD.

## 2. Results

### 2.1. Heat Stress Induced ER Stress and the UPR in Drosophila

We previously showed that heat exposure caused a disturbance in proteostasis not only in the cytosol but also in the ER, and it subsequently induced UPR signaling in mouse embryo fibroblasts (MEFs) [22]. Although several studies showed the hyperthermal effect on *Drosophila* and mice [30,31], there has been no research on how heat stress is correlated with ER stress in vivo. For this purpose, we incubated *Drosophila* at 37 °C for 10 min daily and monitored the survival ratio over the treatment period. The viability of heat-treated flies was significantly decreased around 20 days after the heat treatment, but there were no changes in the survival rate in the flies incubated at 25 °C (Figure 1a). Next, we investigated whether ER stress was induced by heat exposure by determining the levels of UPR proteins. We found that heat exposure significantly elevated the levels of eIF2α phosphorylation in *Drosophila* whole bodies (Figure 1b). We also found that the amount of HSP70 was significantly increased in chronic heat-treated flies (Figure 1b), suggesting that those flies experienced heat shock. Next, we examined the expression levels of the UPR genes, including the total form of *XBP1* (*tXBP1)*, the spliced form of *XBP1* (*sXBP1)*, *ATF6*, *4E-BP*, *PFK*, and *TPI*, in *Drosophila* whole bodies following heat exposure. The amount of *sXBP1* and the expression of *4E-BP*, which are a target of the eIF2α phosphorylation and the ATF4 signaling pathway in the UPR system, were significantly increased after heat exposure (Figure 1c). These data indicated that heat exposure decreased the viability of *Drosophila*, and there was concomitant induction of disturbed ER proteostasis and the UPR signaling pathway in *Drosophila* whole bodies.

### 2.2. Neuronal Cells Were More Susceptible to Heat Stress in Drosophila

Although there was significant induction of the UPR upon heat stress, the induction rate was not dramatic in the whole body of flies. Thus, we hypothesized that there might be more or less susceptible organs within the *Drosophila* body upon heat stress. Interestingly, it has been reported in various studies that neuronal cells are the most susceptible to heat stress and consequently are expected to be the first to undergo dysfunction in vivo [32,33]. A previous study also reported the relationship between heat stress and cognitive function in mice [4]. Furthermore, several studies have shown that heat exposure can lead to neuronal loss, neurological defects, stroke, and neural circuit modification [7,27,32,34]. Therefore, we investigated whether neuronal cells in flies might be more susceptible to hyperthermia-mediated ER stress conditions. First, we checked the mRNA levels of the UPR genes in the heads of flies following heat exposure for 15 and 25 days. Splicing of *XBP1* was significantly increased and the expression of the *4E-BP*, *PFK*, and *TPI* genes, which are targets of the eIF2α signaling pathway in the UPR system, were also increased in the fly heads during chronic heat exposure (Figure 2). Importantly, the induction rate in the head was more dramatic than that of the whole body. This result indicated that chronic heat stress caused ER stress and subsequent induction of the UPR signaling pathway in *Drosophila* heads.

### 2.3. PERK Phosphorylated eIF2α in Drosophila Neurons upon Heat Stress

It has been reported that eIF2α phosphorylation and subsequent attenuation of protein synthesis are critical in the protection of cells from heat-stress-mediated cellular dysfunction and/or death [22]. To further understand the significance of eIF2α phosphorylation under chronic hyperthermal conditions in *Drosophila* brain in an in vivo model, we analyzed the level of eIF2α phosphorylation in head extracts following chronic heat exposure. The level of phosphorylated eIF2α was dramatically increased in head extracts after chronic heat treatment (Figure 3a). This result indicated that chronic heat exposure may activate ER stress in *Drosophila* brain. The phosphorylation of eIF2α is mediated by four different kinases, including PERK, PKR, GCN2, and HRI. Since PERK is reported to be important to sense and transfer the UPR signal upon heat stress [35,36], we examined whether PERK was responsible for the phosphorylation of eIF2α in response to chronic heat exposure in the brain, especially in the neuronal cells. For this purpose, we employed flies with a knockdown of *PERK* under the neuron-specific *elav-Gal4* driver (*UAS-PERK RNAi*). We found that the mRNA amount of *PERK* in head extracts from *UAS-PERK RNAi* was significantly decreased compared with *elav-Gal 4 Drosophila* (Appendix A). The level of eIF2α phosphorylation was not changed in head extracts of *PERK* knockdown flies with chronic heat exposure (Figure 3a), suggesting that chronic heat-induced eIF2α phosphorylation was mediated by PERK in *Drosophila* neuronal cells. Furthermore, we also examined the expression of HSP70 in heads of flies after chronic heat exposure. The expression of HSP70 was increased during exposure to high temperatures in heads of both *elav-Gal4* and *UAS-PERK RNAi Drosophila* (Figure 3b). The expression levels of phosphorylated eIF2α target genes, including *4E-BP*, *TPI*, and *PFK*, were significantly enhanced upon heat stress in wild-type but not in *PERK* knockdown fly heads (Figure 3c). NRF2 is also known to be phosphorylated by activated PERK and important for cell survival during stress [37]. Previous studies also showed that the NRF2 pathway and its target genes, such as *GSTD1*, *HO-1,* and *HSP70*, are increased during heat stress [38,39,40,41]. To investigate the possible role of PERK in activating the NRF2 pathway during heat stress, we checked the expression levels of NRF2 target genes in *Drosophila* head after heat stress. The protein amount of HSP70 (Figure 3b) as well as mRNA amount of *GSTD1* and *HO-1* (Figure 3d) were significantly increased during heat stress in *elav-Gal4 Drosophila*. In contrast with the results of eIF2α target genes, however, there were no significant differences in mRNA levels (*HO-1*) or protein levels (HSP70) between *elav-Gal4* and *UAS-PERK RNAi Drosophila* upon heat stress (Figure 3b,d). The expression level of *GSTD1* was more induced in *UAS-PERK RNAi Drosophila* compared with that in *elav-Gal4 Drosophila* upon heat stress. These results strongly indicate that NRF2 activation upon heat stress in *Drosophila* was not PERK dependent. Although *KEAP1* expression is known to be induced by oxidative-stress-mediated NRF2 activation [42,43], we observed no induction of *KEAP1* mRNA levels upon heat stress and even decreased expression in *PERK* knockdown flies (Figure 3d), indicating that there might be different mechanisms in response to heat stress.

### 2.4. PERK/eIF2α Phosphorylation Was Required to Protect Drosophila upon Heat Stress

We showed that heat stress shortened *Drosophila* life span and eIF2α phosphorylation was more obvious in the brain (Figure 1 and Figure 2). Based on these observations, we hypothesized that the shortened life span of the flies upon heat stress might be correlated with higher susceptibility of the neuronal cells in the brain. First, we examined the viability of the *elav-Gal4* and *UAS-PERK RNAi Drosophila* over the heat-treatment period. The life span of *elav-Gal4* decreased around 26 days after heat exposure and all flies died around 35 days after heat treatment (Figure 4). Interestingly, the life span of neuronal-specific *PERK* knockdown flies (*UAS-PERK RNAi*) decreased around 12 days after heat exposure and was dramatically shorter than that of heat-treated control flies (*elav-Gal4*) (Figure 4). Thus, these results suggest that eIF2α phosphorylation by PERK was required for neuroprotection from chronic heat stress in *Drosophila*.

### 2.5. Attenuation of Protein Synthesis by eIF2α Phosphorylation was Required to Protect Drosophila Brain Tissue from Heat Stress

As an initiation factor, eIF2α phosphorylation and dephosphorylation are critical steps to control protein synthesis in cells [35,44]. It is generally accepted that decreased protein synthesis is critical to protect cells from stress [45,46]. Importantly, we showed that translational attenuation upon heat stress was crucial to protect cells [22]. Based on these results, we hypothesized that the attenuation of protein synthesis by PERK-mediated eIF2α phosphorylation might be required to protect neuronal cells against chronic heat stress in vivo. To determine translation rates in fly heads after chronic heat stress, we performed Surface Sensing of Translation (SUnSET) that employed an anti-puromycin antibody for the immunological detection of puromycin-labeled newly synthesized peptides [47]. We found that amounts of puromycin-incorporated peptides were markedly decreased in heads of *elav-Gal4 Drosophila* after exposure to chronic heat. However, neuron-specific *PERK* knockdown flies did not show a significant decrease in global protein synthesis (Figure 5). These results strongly suggested that attenuation of global protein synthesis by eIF2α phosphorylation was important for neuronal cell survival in the brain during chronic heat stress.

### 2.6. PERK Downregulation in DA Neurons Impaired Locomotive Activity and Dopaminergic Neurons under Chronic Heat Stress

It was reported that thermal stress induces cognitive impairment, including memory loss, in animals and humans [4,27,48]. In addition, repeated heat exposure was closely associated with increased incidence of PD phenotypes in an animal model [5]. Although the effects of heat stress have long been studied, the exact molecular mechanisms by which chronic heat stress affects locomotor behavior and DA neuron survival are not well understood. These observations prompted us to test whether chronic heat stress might be a potential risk factor for PD. For this hypothesis, we examined whether chronic heat-exposed flies exhibited an effect on locomotor activity using a negative geotaxis assay. Although locomotive activity decreased over the experimental time, there was no significant difference in climbing activity in the presence of heat exposure in *Drosophila* (*TH-Gal4*) wild-type controls (Figure 6a). We showed that attenuated protein synthesis by eIF2α phosphorylation was required for protection of cells from heat stress [22]. To determine whether the effect of chronic heat stress in DA neurons was associated with PERK-mediated eIF2α phosphorylation, we downregulated *PERK* specifically in DA neurons of adult flies using a *TH-Gal4* driver and monitored motor activity. The locomotive activity in *PERK*-downregulated flies (*UAS-PERK RNAi*) decreased more dramatically (~40% activity around 25 days) than that of control flies (~70% activity around 25 days) (*TH-Gal4*) (Figure 6a), suggesting the PERK/eIF2α phosphorylation signaling pathway was crucial in the protection of DA neuron-specific toxicity induced by chronic heat exposure. Interestingly, the locomotive activity in *UAS-PERK RNAi* was more significantly reduced around 25 days compared with that of *TH-Gal4* in the absence of heat exposure (Figure 6a).

A hallmark of PD pathology is DA neuron loss, which has been linked to motor impairment in PD animal models and patients [49]. To investigate whether chronic heat stress induced DA neuron loss, fly brains exposed to chronic heat for 25 days were dissected and immunostained for the TH enzyme. Six DA neuron clusters are present in the adult fly brain [50]. Various studies on *Drosophila* have shown that especially DA neurons in the proto-cerebral posterior lateral 1 (PPL1) cluster degenerate in PD models [50,51]. Therefore, we measured the number of DA neurons in the PPL1 region of adult brains. After exposure to chronic heat, there was a significant decrease in the number of DA neurons in the PPL1 region in comparison with control flies without heat (Figure 6b,c), suggesting that chronic heat stress could lead to DA neuron loss in *Drosophila*. In addition, to determine whether PERK could affect DA neuron degeneration, we counted the number of DA neurons in the PPL1 region of *PERK*-downregulated flies. As expected, significantly enhanced DA neuron loss in the PPL1 region was observed in DA neuron-specific *PERK*-downregulated flies upon heat stress (Figure 6b,c). These results indicated that eIF2α phosphorylation mediated by PERK may protect DA neurons against chronic heat stress in *Drosophila*. Interestingly, *PERK*-downregulated flies (*UAS-PERK RNAi*) showed more DA neuron loss in the absence of heat exposure, suggesting the proper function of eIF2α phosphorylation and its subsequent signaling might be crucial in DA neuron survival.

## 3. Discussion

Our present study showed that long-term and mild heat exposure led to a shortened *Drosophila* life span with a concomitant induction of ER stress and the UPR. We also found that UPR signaling was induced much higher in the brain tissues than other parts of the body, suggesting that *Drosophila* brains were more susceptible to heat-mediated ER stress. Moreover, we showed that PERK-mediated eIF2α phosphorylation and subsequent attenuation of protein synthesis in the neurons of *Drosophila* were important for neuronal survival upon heat stress. When the PERK/eIF2α phosphorylation signaling pathway was impaired specifically in the neurons of *Drosophila*, it resulted in a shortened life span with decreased locomotive activity and loss of DA neurons. These observations are consistent with previous studies showing that various stresses, including hyperthermia conditions, cause phosphorylation of eIF2α, which leads to inhibition of global translation and improved cell survival during stress [22,52]

It has been suggested that hyperthermal stress might be implicated in the pathogenesis of neurodegenerative diseases [53]. Exposure to high temperatures resulted in memory-reducing effects in humans and caused cell death in the brain [6,7,9,27]. Recently, it was also shown that heat exposure impairs motor function, dopamine level, and cognitive ability in mice [4,5]. In the current study, we showed that chronic heat exposure recapitulated a similar phenotype of PD, including locomotor dysfunction and DA neuron loss, in a *Drosophila* model. These results suggest that chronic heat exposure might be a potential environmental factor that contributes to the onset of sporadic PD. Although the physiological effects of heat stress have been studied, little is known about the precise mechanism of chronic heat exposure that can lead to neurodegeneration. Our study showed that flies with genetically downregulated *PERK* in DA neurons displayed dramatically decreased motor activity and loss of DA neurons compared with control flies upon chronic heat stress. Besides eIF2α phosphorylation, the NRF2 pathway is also known to be activated by PERK in response to stresses [37,54,55]. Although NRF2 target genes were induced in wild-type fly heads upon heat stress, we did not observe attenuated induction in *PERK* knockdown flies, suggesting that NRF2 activation during heat stress may not be PERK dependent in our system. Although several studies have shown that PERK is involved in NRF2 pathway activation during stresses in mammals [37,54,55], it remained undetermined whether NRF2 activation by PERK also exists in *Drosophila* [56]. Taken together, all the results suggested that PERK-mediated eIF2α phosphorylation upon heat stress might contribute to DA neuron survival in *Drosophila*.

Attenuation of global protein synthesis is known to be a defensive system to protect cells from various types of unfavorable and stressful conditions. Four different kinases, including PERK, PKR, GCN2, and HRI, are activated in response to stresses [57]. Among them, PERK was shown in our previous study to be involved in heat-mediated ER stress and eIF2α phosphorylation. When eIF2α phosphorylation was genetically or chemically inhibited, the cells were more susceptible to heat stress [22]. Consistently, we found in the current study that the flies with genetically knocked down *PERK* in the neuron cells displayed increased susceptibility to heat stress with shortened life span and decreased locomotive activity. There was a dramatically reduced amount of eIF2α phosphorylation with no translational attenuation in the flies with genetically knocked down *PERK*, suggesting that the PERK/eIF2α phosphorylation signaling pathway should be properly working to protect neuronal cells of flies upon heat stress. Surprisingly, we discovered that genetic *PERK* downregulation led to accelerated PD-like phenotypes in chronic heat-exposed *Drosophila*. These data might suggest that PERK/eIF2α phosphorylation signaling was required for heat resistance in DA neurons of *Drosophila*.

Taken together, our findings provide evidence for the contribution of the PERK/eIF2α phosphorylation signaling pathway as a critical mediator of neuronal survival in PD-related neurodegeneration induced by chronic heat exposure. Therefore, the promotion of eIF2α phosphorylation in the brain may be helpful in improving chronic heat-induced PD pathogenesis.

## 4. Materials and Methods

### 4.1. Drosophila stock

*elav-Gal4* (pan-neuronal driver) and *TH-Gal4* (dopamine-neuron-specific neuron) were obtained from the Bloomington *Drosophila* Stock Center. The *PERK* RNAi (v16427) line was obtained from the Vienna *Drosophila* Resource Center. *W^1118^* flies were used as a control. All the stock flies were raised at 25 °C on standard food and crossed using a standard procedure.

### 4.2. Heat Stress Treatment in Flies

Twenty male flies of each genotype were moved into empty vials. The vials were incubated inside a water bath at 37 °C for 10 min. This experiment was done daily at the same time for 25 days.

### 4.3. Life Span Assay

Twenty male flies of each genotype were raised and maintained in different vials at 25 °C. All groups of flies were moved to fresh vials every other day and the number of dead flies was recorded.

### 4.4. Western Blot Analysis 

Twenty-five-day heat-treated male fly whole bodies or heads were homogenized in 4× lithium dodecyl sulfate (LDS) loading buffer and 10× sample reducing agent (Thermo Fisher Scientific, Waltham, MA, USA) for Western blot analysis. To separate the total protein extract, 4–12% gradient SDS-PAGE (Invitrogen, Carlsbad, CA, USA) was used, which was transferred to polyvinylidene difluoride membranes (Millipore, Burlington, MA, USA). The primary antibodies used were as follows: total eIF2α (1:200; Abcam, Cambridge, UK, ab26197-100), phosphor-eIF2α (1:1000; Cell Signaling, Danvers, MA, USA, 9721S), β-actin (1:5000; Cell Signaling, Danvers, MA, USA, 4967S), or HSP70 (1:1000; Enzo Life Sciences, Farmingdale, NY, USA, SPA-822). Secondary antibodies used were as follows: goat anti-rabbit IgG horseradish peroxidase (HRP) and goat anti-mouse IgG HRP conjugate (1:2000; Millipore, Burlington, MA, USA, AP307P, AP308P). Proteins were detected using an ECL-Plus kit (Thermo Fisher Scientific, Waltham, MA, USA). 

### 4.5. RNA Extraction and Real-Time PCR Analysis

Total RNA was extracted from fly whole bodies and heads using TRIzol (Sigma, St. Louis, MO, USA) and cDNA was synthesized using an iScript cDNA synthesis kit (Bio-Rad, Hercules, CA, USA, BR170-8891). The relative amounts of mRNAs were calculated from the comparative threshold cycle (Ct) values relative to Rpl32 rRNA using a CFX96 real-time PCR detection system (Bio-Rad, Hercules, CA, USA, 184-5384) with SYBR green reagent (Enzynomics, Daejeon, Korea, RT500M), according to the manufacturer’s instructions. Real-time primer sequences used in this study were as follows: *Rpl32* (5′-CGGATCGATATGCTAAGCTGT-3′; 5′-GCGCTTGTTCGATCCGTA-3′) *tXBP1* (5′-TCTAACCTGGGAGGAGAAAG-3′; 5′-GTCCAGCTTGTGGTTCTTG-3′), *sXBP1* (5′-CCGAACTGAAGCAGCAACAGC-3′; 5′-GTATACCCTGCGGCAGATCC-3′), *ATF6* (5′-AACGTAATTCCACGGAAGCCCAACA-3′; 5′-GCGACGGTAGCTTGATTTCTAGAGCC-3′) PEK 5′- TACTAGGTCCAGTGGTGC-3′; 5′- GCTTGTCCAGGTGGGAAGCTA-3′ [58], *4E-BP* (5′-GCTAAGATGTCCGCTTCACC-3′; 5′-CCTCCAGGAGTGGTGGAGTA-3′) [59], *PFK* (5′-CTGCAGCAGGATGTCTACCA-3′; 5′-GTCGATGTTCGCCTTGATCT-3′), *TPI* (5′-GACTGGAAGAACGTGGTGGT-3′; 5′-CGTTGATGATGTCCACGAAC-3′) [60]. *GSTD1* (5′-GGCCGCCTTCGAGTTCCTGA-3′; 5′- CGGTTGCCACCAGGGCAATG-3′), KEAP1 (5′- TGGCCAGCGTGGAGTGCTAC-3′; 5′- TTGCAGCAACACCCGCTCCA-3′) [61], HO-1 (5′- ACCATTTGCCCGCCGGGATG-3′; 5′- AGTGCGAGGGCCAGCTTCCT-3′) [62].

### 4.6. Puromycin Incorporation Assay

To measure nascent protein synthesis, 10 25-day heat-treated male flies were homogenized in 80 μL of cold solubilization buffer (15 mM Tris-HCl, 300 mM NaCl, 15 mM MgCl_2_, 2 mM DTT, 1% Triton X-100, and 12.5 μL/mL RNase In; pH 7.5) supplemented with 100 μM puromycin. Lysates were incubated at 4 °C for 5 min and mixed with 40 μL of 4× LDS loading buffer (Thermo Fisher Scientific, Waltham, MA, USA). Homogenized samples were then boiled for 8 min and centrifuged at 20,000× *g* for 30 min. To separate the supernatant, 4–12% gradient SDS-PAGE (Invitrogen, Carlsbad, CA, USA) was used, which was transferred to polyvinylidene difluoride membranes (Millipore, Burlington, MA, USA). The membrane was incubated with mouse anti-puromycin antibody (1:10000; Millipore, Burlington, MA, USA, MABE343) overnight and was detected with HRP-conjugated mouse secondary antibodies (1:10000; Millipore, Burlington, MA, USA, AP308P). Detection was carried out using an ECL-Plus kit (Thermo Fisher Scientific, Waltham, MA, USA).

### 4.7. Locomotive Activity Assay

The locomotive activity assay used the characteristics of the flies moving against the force of gravity (geotaxis). Twenty male flies of each genotype were placed in a column vial. Flies were tapped to the bottom of the column and the locomotive activity was counted by measuring the distance and speed of the flies to climb to the top of vials. This assay was repeated several times for each genotype of flies.

### 4.8. Immunohistochemistry

Twenty-five-day heat-treated male fly brains of each genotype were dissected and incubated with 4% formaldehyde in fixative buffer (100 mM PIPES, 1 mM ethylene glycol tetra acetic acid, 1% Triton X-100, and 2 mM MgSO_4_; pH 6.9) for 30 min, permeabilized with 1% Triton X-100 in washing buffer (50 mM Tris-HCl, 150 mM NaCl, 0.1% Triton X-100, and 0.5 mg/mL BSA; pH 6.8) for 10 min, and blocked with 10 mg/mL BSA in washing buffer at 4 °C overnight. The brains were then stained overnight at 4 °C with mouse anti-TH antibody (1:100; Immuno Star, Hudson, WI, USA, #22941) and anti-mouse Cy3-conjugated secondary antibodies (1:200; Jackson Immunoresearch, West Grove, PA, USA, A10521) were used for identification. All images were analyzed using a DE/LSM710 NLO Carl Zeiss confocal microscope (Oberkochen, Germany).

## Figures and Tables

**Figure 1 ijms-21-00845-f001:**
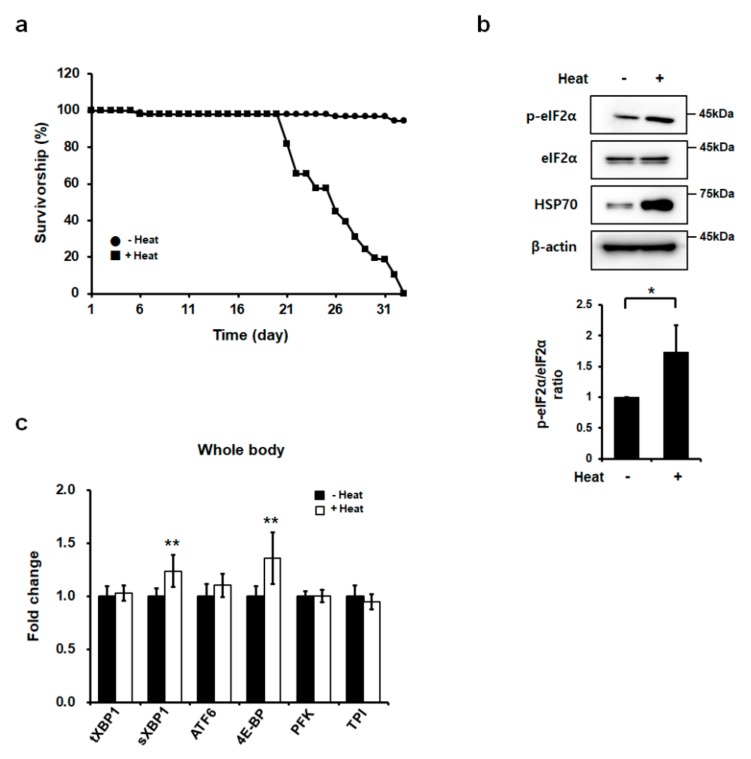
Chronic heat exposure led to decreased life span and increased endoplasmic reticulum (ER) stress in *Drosophila*. (**a**) Chronic heat stress decreased *Drosophila* life span. Survival curves of heat-treated flies (■) and control flies (●) are shown. A total of 150 males were assayed for each genotype. Flies from each experiment were subjected to survival assays at 25 °C. (**b**) Protein expression levels in whole bodies of chronic 25-day heat-treated flies. β-actin was used as a loading control. (**c**) ER stress gene expression levels in whole bodies of chronic 25-day heat-treated flies. Quantitative RT-PCR was performed using total RNA extracted from whole bodies of heat-treated flies. Error bars represent mean ± standard deviation of three independent experiments. The experimental significance was determined using a one-way ANOVA (**p* < 0.05; ***p* < 0.01).

**Figure 2 ijms-21-00845-f002:**
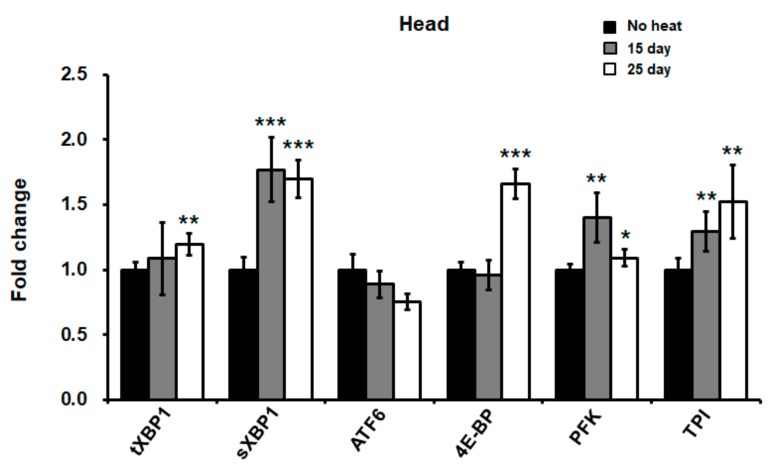
Chronic heat exposure induced ER stress mediated by the ATF4 pathway in *Drosophila* heads. The expression levels of ER stress genes in heads of chronic 15- and 25-day heat-treated flies. Quantitative RT-PCR was performed using total RNA extracted from heads of heat-treated flies. Error bars represent ± standard deviation from three independent experiments. Statistical significance was determined using a one-way ANOVA (**p* < 0.05; ***p* < 0.01; ****p* < 0.001).

**Figure 3 ijms-21-00845-f003:**
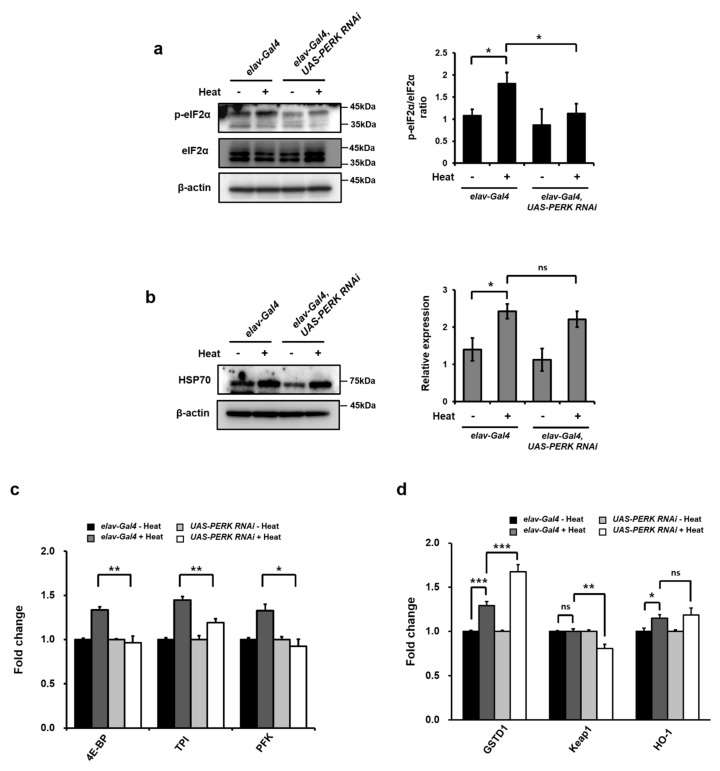
PKR-like eukaryotic initiation factor 2α kinase (PERK) was responsible for eIF2α phosphorylation in response to chronic heat exposure in *Drosophila* neurons. (**a**) Protein levels of p-eIF2α in 25-day heat-treated fly heads. Chronic heat stress for 25 days resulted in increased p-eIF2α in *elav-Gal4* flies. β-actin was used as a loading control. (**b**) HSP70 levels in 25-day heat-treated fly heads. β-actin was used as a loading control. The expression levels of ATF4 target gene (**c**) and NRF2 target gene (**d**) in indicated flies. Quantitative RT-PCR was performed using total RNA extracted from heads of heat-treated flies. Error bars represent mean ± standard deviation of three independent experiments. The experimental significance was determined using a one-way ANOVA (**p* < 0.05; ***p* < 0.01; ns, not significant).

**Figure 4 ijms-21-00845-f004:**
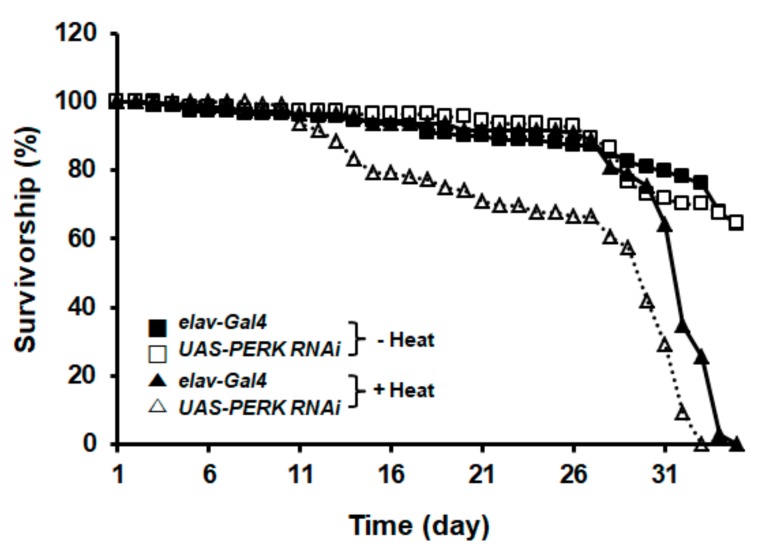
PERK/eIF2α phosphorylation was required for protection from chronic heat-stress-induced neuronal toxicity. Survival curves of *PERK* knockdown flies (Δ) and controls (▲) upon chronic heat exposure showed reduced survival in *PERK* knockdown flies. *PERK* knockdown flies (□) and control flies (■) showed normal longevity without heat stress during the time course of the experiment. A total of 150 males were assayed for each genotype. Flies from each genotype were subjected to survival assays at 25 °C.

**Figure 5 ijms-21-00845-f005:**
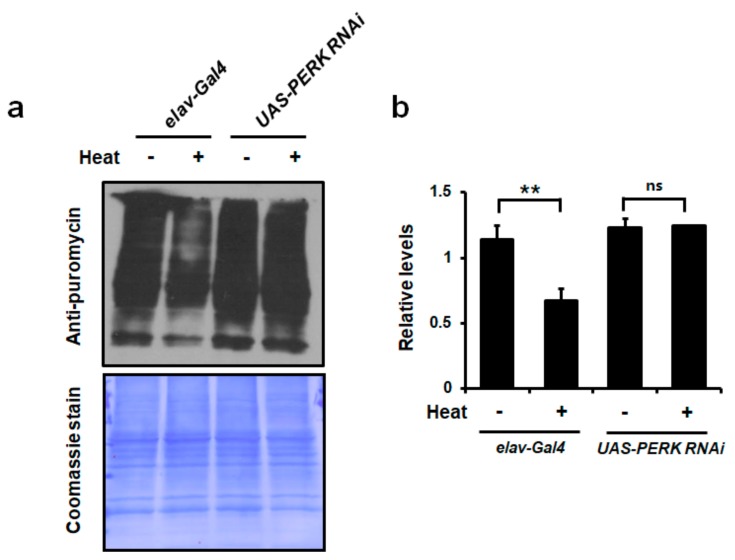
Translation rate in the brain was attenuated during chronic heat stress. (**a**) Puromycin incorporation assay results in 25-day heat-treated fly heads. Chronic heat stress for 25 days resulted in the decrease of global protein translation in *elav-Gal4* flies. Coomassie staining is shown as a loading control. (**b**) Quantification of (**a**) error bars represents the mean ± standard deviation of three independent experiments. The experimental significance was determined using a one-way ANOVA (***p* < 0.01; ns, not significant).

**Figure 6 ijms-21-00845-f006:**
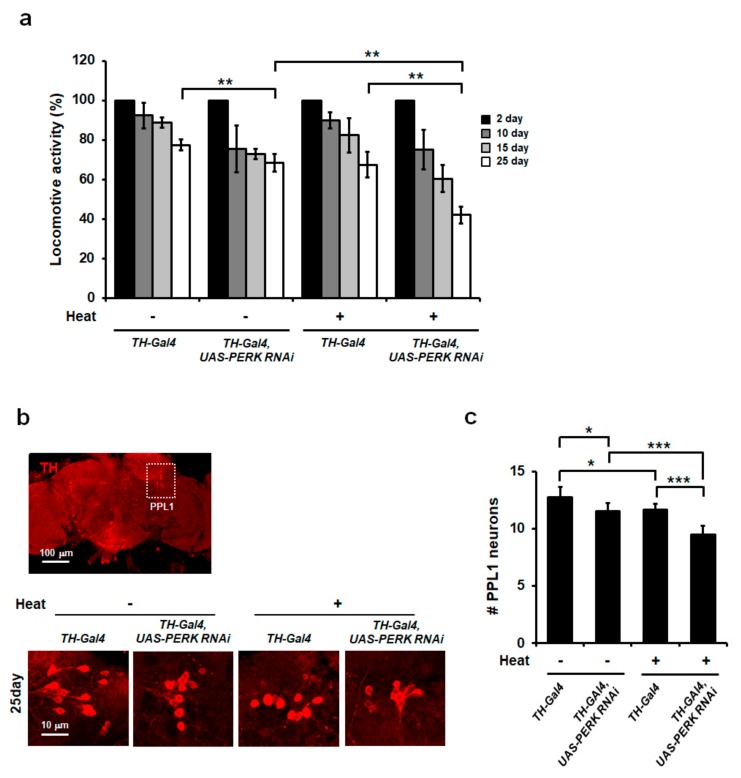
Loss of dopaminergic (DA) neurons of *PERK* RNAi-expressing flies under chronic heat stress. (**a**) Percentage of locomotive activity of control and dopamine neuron-specific *PERK* RNAi overexpressing flies. Overexpression of *PERK* RNAi in dopaminergic neuronal cells reduced locomotive activity and this genotype of flies displayed severely defective locomotive activity under heat stress conditions. Error bars represent ± standard deviation of three independent experiments. Statistical testing was evaluated by a two-way ANOVA with a Tukey’s multiple comparison test (** *p* < 0.01). (**b**) Tyrosine hydroxylase (TH) immunohistochemistry in brains. Brains dissected from 25-day heat-treated flies from wild-type (*TH-Gal4*) and dopamine-neuron-specific *PERK* RNAi overexpression with and without heat stress were stained with anti-TH antibody (red). (**c**) The quantification of dopamine neurons marked by TH-positive staining. The number of TH-positive neurons in the brain was significantly decreased in dopamine-neuron-specific *PERK* RNAi overexpressing flies in heat stress conditions (*n* = 8 brains per genotypes). Statistical significance was determined using a two-way ANOVA with a Tukey’s multiple comparison test (* *p* < 0.05; *** *p* < 0.001).

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
