# Peer review of "PERK-Mediated eIF2α Phosphorylation Contributes to The Protection of Dopaminergic Neurons from Chronic Heat Stress in Drosophila"

_ijms, 2020, doi:10.3390/ijms21030845_

Round 1

Reviewer 1 Report

The UPR as result of ER stress leads to phosphorylation of eIF2α. The latter event has been shown be essential is several disease models including neurodegenerative disorders. Here, Elvira et al show that chronic heat exposure to flies initiate ER stress, and subsequently increased eIF2α phosphorylation. The authors claim that eIF2α offers dopaminergic neurons survival from chronic heat stress in flies. However, my following concerns need to be addressed before the manuscript is deemed suitable for a publication in IJMS.

In general, ER stress precedes cells death. Here authors show that the increased p-eIF2α in 25-day heat exposed flies (Figure 1b and c). Apparently, the survival of flies drops around 20 day treatment. What happens at early phase of the heat treatment?. A time course analysis (similar to Figure 6A) of biochemical changes in UPR markers is important to support the conclusion. Throughout the manuscript the authors claim that PERK is involved in the ER stress response pathway, which is largely expected. But never in the manuscript authors show PERK protein is indeed depleted by RNAi. A western blot of PERK protein from fly head extract needs to shown for the knockdown experiments. ER stress affects virtually all cell types and all types of neurons. Are only dopaminergic neurons affected by chronic heat stress? How about other neuronal cell types? TH immunohistochemistry should be complemented by TH western blot.

Author Response

Response to Reviewer 1 Comments

The UPR as result of ER stress leads to phosphorylation of eIF2α. The latter event has been shown be essential is several disease models including neurodegenerative disorders. Here, Elvira et al show that chronic heat exposure to flies initiate ER stress, and subsequently increased eIF2α phosphorylation. The authors claim that eIF2α offers dopaminergic neurons survival from chronic heat stress in flies. However, my following concerns need to be addressed before the manuscript is deemed suitable for a publication in IJMS.

Point 1: In general, ER stress precedes cells death. Here authors show that the increased p-eIF2α in 25-day heat exposed flies (Figure 1b and c). Apparently, the survival of flies drops around 20-day treatment. What happens at early phase of the heat treatment? A time course analysis (similar to Figure 6A) of biochemical changes in UPR markers is important to support the conclusion.

Response 1: We appreciate the insightful comment to improve the manuscript. As the reviewer suggested, we newly analyzed the expression levels of UPR marker genes in extracts of fly heads on 15 days following heat exposure. This new information is now added in Figure 2 and line 135 of revised manuscript.

Point 2: Throughout the manuscript the authors claim that PERK is involved in the ER stress response pathway, which is largely expected. But never in the manuscript authors show PERK protein is indeed depleted by RNAi. A western blot of PERK protein from fly head extract needs to be shown for the knockdown experiments.

Response 2: We appreciate the insightful comment to improve the manuscript. As the reviewer suggested, we analyzed the expression levels of Perk in heads of flies. We found significantly decreased amount of Perk mRNA in head extracts from the UAS-PERK RNAi flies compared to that from elav-Gal4, suggesting that Perk gene expression is efficiently knock-downed. This new information was added in the Supplementary Figure S1 (Appendix A) and lines between 153-155 in the revised manuscript. Unfortunately, we could not find suitable antibody that cross-reacts with Drosophila PERK and we could not analyze the content of PERK protein in fly extract using western blot analysis. However, we observed increased amount of phosphorylated eIF2a in control but not in knock-downed flies upon heat stress (Figure 3a), suggesting that Perk gene expression is efficiently knock-downed. We hope the reviewer understand the technical difficulties.

Point 3: ER stress affects virtually all cell types and all types of neurons. Are only dopaminergic neurons affected by chronic heat stress? How about other neuronal cell types?

Response 3: We agree with reviewer and appreciate the constructive and insightful comments. It is known that dopaminergic (DA) neurons are particularly vulnerable to stress than other types of neurons due to the presence of ROS generating molecules including dopamine, iron, as well as low levels of antioxidants. Therefore, we focused on the effect of heat stress on DA neurons first. However, we are also wondering whether ER stress affects the survival of other types of neurons. A detailed study is currently under way in our group.

Point 4: TH immunohistochemistry should be complemented by TH western blot.

Response 4: Thank you for your suggestions. We have tried most of commercially available TH antibodies. Although most TH antibodies worked for immunofluorescent experiment, we could not find suitable TH antibody for western blot analysis. Therefore, we couldn’t check the expression level of TH in each genotype of flies using western blot analysis. We hope the reviewer to understand technical limitation in this study.

Reviewer 2 Report

Despite the fact that the manuscript is written clearly and the methods used in the work are adequate, the reviewer cannot recommend this work for publication at current condition for the following reasons:

First and foremost, the conclusions presented by the authors do not follow from the results obtained. Homogenized tissue from whole fly head was used to obtain protein and RNA extracts for Western blot analysis, real-time-PCR analysis as well as SUnSET method. However, the observed changes in the expression of ER stress genes, eif2a phosphorylation and the level of translation are considered by the authors as changes in neuronal cells. A whole head contains not only neurons, does it? Moreover, they conclude that the same processes occur in DA neurons. Why was Student t-test used to assess the loss of DA neurons , while a two-way ANOVA multi-comparison test was used to evaluate locomotor activity? ANOVA test did not show significant cell loss?

Thus, only conclusions based on data showing locomotor activity are relevant.

Authors ignored in Introduction to mention multiple publications demonstrating alternative opinion and data on mammals regarding role of translation block induced by prolonged eif2a phosphorylation. They suggest that the PERK downstream target, p-eIF2a, reduces the burden on the cell by temporal inhibition of translation. However, if this activation is prolonged, it activates apoptotic neuronal cell death through up-regulation of ATF4→ CHOP and, most important, significantly reduce varies vital synaptic proteins. For example:

Alzheimer’s disease. PERK-/- reduces p-eIF2α  and deficit in protein synthesis, synaptic plasticity, and spatial memory in APP/PS1 AD model mice (Tao Ma et al., 2013).

Prion disease. Over-expression of GADD34  and RNAi delivery reduce p-eIF2α, restore vital translation rates during prion disease, rescue synaptic deficits and neuronal loss, and thereby significantly increase survival (Moreno et al., 2012)

Parkinson’s disease. Targeting of PERK by inhibitor GSK2606414 results in ER stress inhibition, improving motor performance and increasing dopamine levels and the expression of synaptic proteins. (Mercado et al, 2018).

Use of salubrinal (inhibits p-eIF2a dephosphorylation) transgenic A53T mouse model of PD (Colla et al. 2012) resulted in delay of motor behavior deficit and the rescue of Golgi morphology. However, these authors themselves noted that “salubrinal does not increase the survival of DA neurons destined for cell death.”

Author Response

Response to Reviewer 2 Comments

Despite the fact that the manuscript is written clearly and the methods used in the work are adequate, the reviewer cannot recommend this work for publication at current condition for the following reasons:

Point 1: First and foremost, the conclusions presented by the authors do not follow from the results obtained. Homogenized tissue from whole fly head was used to obtain protein and RNA extracts for Western blot analysis, real-time-PCR analysis as well as SUnSET method. However, the observed changes in the expression of ER stress genes, eif2a phosphorylation and the level of translation are considered by the authors as changes in neuronal cells. A whole head contains not only neurons, does it?

Response 1: We appreciate the reviewer’s constructive critique. The Gal4/UAS system is commonly used in Drosophila to drive expression of a specific gene. RNAi for a specific gene is expressed under UAS and when crossed with a Gal4 driver, the relevant protein can be knocked down in specific tissue of interest. In this study, we expressed PERK RNAi in neuronal cells using pan-neuronal driver, elav-Gal4 and analyzed defective phenotypes in PERK RNAi-expressing flies under heat exposure using head extracts. Therefore, although we cannot completely exclude the possibility that defective phenotypes might be due to non-neuronal cells, our studies provide that PERK is required for phosphorylation of eIF2α in response to chronic heat-induced ER stress activation in neuronal cells.

Point 2: Moreover, they conclude that the same processes occur in DA neurons. Why was Student t-test used to assess the loss of DA neurons, while a two-way ANOVA multi-comparison test was used to evaluate locomotor activity? ANOVA test did not show significant cell loss? Thus, only conclusions based on data showing locomotor activity are relevant.

Response 2: Thank you for your valuable comments. The p values of the experimental groups were calculated by comparing the experimental groups to the elav-Gal4 or UAS-PERK RNAi groups using a two-way ANOVA with a Tukey’s multiple-comparison test. We have modified our statement on page 9 in the Figure legends line 258-259 of revised manuscript as follows:

“Statistical significance was determined using a two-way ANOVA with a Tukey’s multiple-comparison test (*p < 0.05; ***p < 0.001).”

Point 3: Authors ignored in Introduction to mention multiple publications demonstrating alternative opinion and data on mammals regarding role of translation block induced by prolonged eif2a phosphorylation. They suggest that the PERK downstream target, p-eIF2a, reduces the burden on the cell by temporal inhibition of translation. However, if this activation is prolonged, it activates apoptotic neuronal cell death through up-regulation of ATF4→ CHOP and, most important, significantly reduce varies vital synaptic proteins.

Response 3: We appreciate the insightful comment to improve the manuscript. As reviewer mentioned, prolonged activation of p-eIF2α will up-regulated ATF4 and CHOP leading to the apoptotic neuronal cell death especially in Parkinson Disease (Mercado et al., 2018, Colla et al., 2012). However, it is of note that the UPR is a defensive system for cells to protect themselves from the stress-induced damage. In case of eIF2α phosphorylation, it is beneficial to the cells by reducing the translation, thus conserving energy source and enhancing translation of selected mRNAs expressions of stress-related protein (Tao et al., 2013, Wulf et al., 2007, Wek et al., 2007). In addition, previous study also showed that the downstream of eIF2α phosphorylation ATF4, also protects neuronal cell death in PD models by maintaining Parkin levels (Sun et al., 2013). We added the explanation about this in the introduction part between lines 70-78 of revised manuscript as follows:

“ER stress, such as phosphorylation of PERK and eIF2α has been known to be activated in PD [20]. The role of this eIF2 in PD might have positive and negative effect on dopamine neuron survival. Previous studies have suggested that inhibiting PERK pathway chemically can increase the motor performance in the mice models [21, 22]. On the other hand, it was reported that eIF2α activation have a protective effect by reducing protein translation, thus conserving energy resources and enhancing translation of selected mRNAs expressions of stress related protein [23-25]. In addition, previous studies also shown that the downstream of eIF2α phosphorylation ATF4, also protects neuronal cell death in PD models by maintaining Parkin levels [26].”

Round 2

Reviewer 1 Report

The authors have adequately addressed my concerns. The manuscript is now suitable for publication in IJMS.

Author Response

Thank you for your comments.

Reviewer 2 Report

Major concern: Authors suggested that PERK mediated sustained phosphorylation of eiF2a resulting in attenuation of translation may protect DA neurons against chronic heat stress. To prove their hypothesis, they used the opposite approach, i.e. genetic downregulation of PERK. They demonstrated that ‘…genetic downregulation of PERK, specifically in dopaminergic (DA) neurons, impaired motor activity and led to DA neuron loss.’  However, PERK is not only kinase known to phosphorylate eiF2a. PERK was also established as a direct activator of the Nrf2 transcription factor and Nrf2 target genes. Moreover, it was shown that ‘…Nrf2 nuclear translocation is independent of eIF2α phosphorylation … and… targeted deletion of Nrf2 reduces cell survival following ER stress’ (Cullian et al 2003). Nrf2, in turn, and its downstream pathways have been shown to play very important role in cell survival during heat shock. Most likely, Nrf2 response to heat stress is significantly diminished in PERKi fly model used by authors.  Since they did not investigate Nrf2, it remains unclear which PERK’s downstream, p-eiF2a or Nrf2 (or both), is responsible for impaired motor activity and DA neuron loss.

Author Response

Response to Reviewer 2 Comments

Point: Authors suggested that PERK mediated sustained phosphorylation of eiF2a resulting in attenuation of translation may protect DA neurons against chronic heat stress. To prove their hypothesis, they used the opposite approach, i.e. genetic downregulation of PERK. They demonstrated that ‘…genetic downregulation of PERK, specifically in dopaminergic (DA) neurons, impaired motor activity and led to DA neuron loss.’  However, PERK is not only kinase known to phosphorylate eiF2a. PERK was also established as a direct activator of the Nrf2 transcription factor and Nrf2 target genes. Most likely, Nrf2 response to heat stress is significantly diminished in PERKi fly model used by authors.  Since they did not investigate Nrf2, it remains unclear which PERK’s downstream, p-eiF2a or Nrf2 (or both), is responsible for impaired motor activity and DA neuron loss.

Response: We appreciate the insightful comment to improve the manuscript. To answer this concern, we checked the expression of NRF2 gene target after heat stress in wildtype and PEK knockdowned Drosophila head. Although eif2a target genes were induced, the expression levels of Nrf2 target gene were not changed upon heat stress in PERK-knockdowned fly head. These results suggest that PERK activation is involved in eIf2a phosphorylation signaling pathway but not in NRF2 signaling pathway upon heat stress. We added this information in Figure 3c-d as well as in revised manuscript in lines 164-174 as follow:

“Since NRF2 is also known to be phosphorylated by activated PERK and important for cell survival during stress [38], we checked the expression levels of NRF2 target genes in Drosophila head after the heat stress. The expression levels of phosphorylated eIF2a target genes including 4ebp, Tpi, and Pfk were significantly enhanced upon heat stress in wild type but not in Perk knockdowned fly heads

(Figure 3c). In contrast, the expression patterns of NRF2 target genes including Gstd1, Keap1, Ho-1 showed no obvious Perk-dependency upon heat stress (Figure 3d).”

In addition, we changed the figure 3 legend in page 7 as follows:

“The expression levels of ATF4 target gene (c) and NRF2 target gene (d) in indicated flies. Quantitative RT-PCR was performed using total RNA extracted from heads of heat-treated flies. Error bars represent mean ± standard deviation of three independent experiments. The experimental significance was determined using a one-way ANOVA (*p < 0.05; **p < 0.01; ns, not significant).”

Round 3

Reviewer 2 Report

The reviewer cannot recommend this manuscript publication for the following reasons:

Major concerns: Multiple publications indicate that heat shock induces the cellular defenses through Nrf2 and its downstream signaling pathways. Although the authors present their new quantitative RT-PCR data showing selective NRF2 target gene expressions, they do not provide convincing evidence that the Nrf2 and its downstreams are not activated in response to heat.

Here is again, use of whole fly head tissue cannot reflect the processes occurring not only in DA cells, but also in neurons in general, because it contains a huge amount of non-specific cells masking the real neuronal response.

Minor concerns: line 73-76:  Ref.25 (Cola et al., 2012) incorrect citing.

Author Response

Response to Reviewer 2 Comments

Point: There is an incorrect citing in line 73-76:  Ref.25 (Cola et al., 2012).

Response: We appreciate kind comment to improve our manuscript. According Reviewer’s suggestion, we checked and changed the citation in line 74-76 (Ref 25) to

“Celardo, I.; Costa, A. C.; Lehmann, S.; Jones, C.; Wood, N.; Mencacci, N. E.; Mallucci, G. R.; Loh, S. H. Y.; Martins, L. M., Mitofusin-mediated ER stress triggers neurodegeneration in pink1/parkin models of Parkinson’s disease. Cell Death & Disease 2016, 7, (6), e2271-e2271.”